# Supportive Care in Pediatric Oncology: Opportunities and Future Directions

**DOI:** 10.3390/cancers15235549

**Published:** 2023-11-23

**Authors:** Jason L. Freedman, Dori M. Beeler, Alison Bowers, Natalie Bradford, Yin Ting Cheung, Maya Davies, L. Lee Dupuis, Caitlin W. Elgarten, Torri M. Jones, Tracey Jubelirer, Tamara P. Miller, Priya Patel, Charles A. Phillips, Hannah R. Wardill, Andrea D. Orsey

**Affiliations:** 1Department of Pediatrics, Perelman School of Medicine, University of Pennsylvania, Philadelphia, PA 19104, USA; 2Division of Oncology, Children’s Hospital of Philadelphia, Philadelphia, PA 19104, USA; 3Department of Supportive Oncology, Levine Cancer Institute, Atrium Health, Charlotte, NC 28204, USA; 4Cancer and Palliative Care Outcomes Centre, Centre for Children’s Health Research, Queensland University of Technology, Brisbane City, QLD 4000, Australia; 5School of Pharmacy, Faculty of Medicine, The Chinese University of Hong Kong, Hong Kong SAR, China; 6Hong Kong Hub of Paediatric Excellence, The Chinese University of Hong Kong, Hong Kong SAR, China; 7School of Biomedicine, The Faculty of Health and Medical Sciences, University of Adelaide, Adelaide, SA 5005, Australia; 8Supportive Oncology Research Group, Precision Cancer Medicine Theme, The South Australian Health and Medical Research Institute, Adelaide, SA 5001, Australia; 9Leslie Dan Faculty of Pharmacy, University of Toronto, Toronto, ON M5S 1A1, Canada; 10Child Health Evaluative Sciences, Research Institute, The Hospital for Sick Children, Toronto, ON M5G 1E8, Canada; 11Department of Pharmacy, The Hospital for Sick Children, Toronto, ON M5G 1E8, Canada; 12Department of Psychiatry, Perelman School of Medicine, University of Pennsylvania, Philadelphia, PA 19104, USA; 13Aflac Cancer & Blood Disorders Center, Children’s Healthcare of Atlanta, Emory University School of Medicine, Atlanta, GA 30322, USA; 14Pediatric Oncology Group of Ontario, Toronto, ON M5G 1V2, Canada; 15Department of Biomedical and Health Informatics, Children’s Hospital of Philadelphia, Philadelphia, PA 19104, USA; 16Center for Cancer and Blood Disorders, Connecticut Children’s Medical Center, Hartford, CT 06106, USA; 17Department of Pediatrics, University of Connecticut School of Medicine, Farmington, CT 06032, USA

**Keywords:** pediatrics, supportive care, clinical practice guidelines, CPG, PRO, palliative care, survivorship, nutrition, equity, pediatric oncology, stem cell transplant

## Abstract

**Simple Summary:**

Supportive care is an essential component of excellent cancer care. It refers to the optimization of outcomes through supportive practices such as infection prevention, nutritional improvements, survivorship, early integration of palliative care, and addressing the psychological effects of cancer therapy. This perspective highlights the future directions and efforts necessary to advance supportive care in order to improve outcomes, survival, and quality of life for pediatric cancer patients. Herein, the authors summarize critical accomplishments and highlight important opportunities to expand research in this field to ensure optimal outcomes for children with cancer and survivors of childhood cancer.

**Abstract:**

The optimization of outcomes for pediatric cancer patients relies on the successful advancement of supportive care to ease the treatment burden and mitigate the long-term impacts of cancer therapy. Advancing pediatric supportive care requires research prioritization as well as the development and implementation of innovations. Like the prevailing theme throughout pediatric oncology, there is a clear need for personalized or precision approaches that are consistent, evidence-based, and guided by clinical practice guidelines. By incorporating technology and datasets, we can address questions which may not be feasible to explore in clinical trials. Now is the time to listen to patients’ voices by using patient-reported outcomes (PROs) to ensure that their contributions and experiences inform clinical care plans. Furthermore, while the extrapolation of knowledge and approaches from adult populations may suffice in the absence of pediatric-specific evidence, there is a critical need to specifically understand and implement elements of general and developmental pediatrics like growth, nutrition, development, and physical activity into care. Increased research funding for pediatric supportive care is critical to address resource availability, equity, and disparities across the globe. Our patients deserve to enjoy healthy, productive lives with optimized and enriched supportive care that spans the spectrum from diagnosis to survivorship.

## 1. Introduction

Due to therapeutic advances, the survival rates for many pediatric cancers are among the highest of all cancer types. With decades of life left to live after their cancer diagnosis, pediatric patients’ quality of life is paramount. Despite this understanding, the global approach to pediatric supportive care and survivorship remains fragmented, and is largely extrapolated from adults, a group in which great advances in supportive care have been made. Experts in supportive care from around the globe are united in advancing supportive care practice, science, delivery, implementation, and standardization. Yet, critical areas of development require further exploration in order to ensure that children with cancer have the best, safest, and most optimized outcomes, as well as the highest possible quality of life (Table 1). Achieving these aims will require the collaboration and harmonization of consortia across the globe, like the International Society of Pediatric Oncology (SIOP), Children’s Oncology Group (COG), and the Multinational Association for Supportive Care in Cancer (MASCC). A collaborative, concerted effort to optimize supportive care will benefit all pediatric cancer patients around the globe.

## 2. Optimizing Outcomes with Clinical Practice Guideline-Consistent Care

Clinical practice guideline (CPG) recommendations are actionable statements that address health questions. They are, by definition, created by expert panels and founded on systematic reviews of the literature [1,2]. Providing CPG-consistent care improves patient outcomes [3], boosts clinician satisfaction with the care they provide [4], and increases the cost-effectiveness of care delivery [5,6]. Thus, pediatric oncology patients, clinicians, and healthcare systems are best served when patients receive CPG-consistent supportive care.

Until 2011, when the first pediatric supportive care CPG on the classification of chemotherapy emetogenicity was published, no published pediatric supportive care CPGs were available [7]. In 2017, pediatric oncology clinicians were encouraged to develop and implement supportive care CPGs [8]. Since then, oncology organizations, including MASCC, that develop, endorse, or facilitate the implementation of pediatric supportive care CPGs, have formed the International Pediatric Oncology supportive care Guideline (iPOG) Network [9]. The iPOG Network houses a repository of guidance developed or endorsed by member organizations.

### 2.1. Clinical Practice Guideline (CPG) Implementation

For CPGs to be successfully implemented, health care professionals and stakeholders at local institutions must prioritize their use and tailor the source CPG recommendations based on institutional resources, priorities, cultures, and values. Many CPG developers signal the certainty that the benefits of a CPG recommendation will outweigh its disadvantages by ranking its strength (strong or conditional/weak) [10]. This can help to guide an institution’s decision to adopt, adapt, or reject a source CPG recommendation, since a strong recommendation should, most often, be adopted and implemented as a policy. In contrast, a conditional recommendation will likely require adaptation to the local context. The evidence that informs many supportive care CPGs reflects the care provided in highly resourced settings. Transparent CPG adaptation to low or moderately resourced settings is, nevertheless, an important way to encourage the provision of evidence-based supportive care for all pediatric patients regardless of where they receive care. Involving and educating stakeholders in the CPG review and adaptation process is crucial to successful CPG implementation. Local CPG implementation then requires the education of stakeholders, including patients and families, and the creation of tools to facilitate CPG-consistent care [11,12]. Focused education regarding key practice changes that result from CPG implementation is especially important.

### 2.2. Addressing Gaps in Evidence Regarding Pediatric Supportive Care

Although the number of pediatric supportive care CPGs has increased in recent years, many important supportive care topics (e.g., tumor lysis syndrome prevention, prophylaxis of viral infections, management of fever in non-neutropenic patients) remain unaddressed [2,13]. The reasons for this are many, ranging from a lack of sufficient published evidence to conduct a meaningful systematic review to resource limitations among CPG developers. When published evidence is absent or sparse, CPG developers may contribute to evidence creation [14]. Pragmatically, it may be reasonable to cautiously generalize from a high-quality CPG on a topic that was created for use in adult patients or in pediatric patients without cancer (e.g., constipation in general pediatric patients) [15]. The adaptation of a CPG for use in patients other than those for whom a CPG was originally designed must be purposeful and include an explanation of why generalization is acceptable. In the absence of CPG recommendations on a topic, expert opinion statements (statements that do not meet the definition of a CPG-derived recommendation or a good practice statement) may be useful [2]. However, the evaluation of patient outcomes following implementation of care based on expert opinion is critical, since it ensures high-quality care and contributes to the evidence base that may inform a future CPG on the topic.

## 3. Empowering Voices in Pediatric Oncology

It is crucial to consider important factors that can improve communication and empower the voices of children, adolescents, and caregivers throughout the entire care journey. While tools like patient-reported outcomes (PROs) can help navigate these complexities, implementing such tools into practice remains challenging.

### 3.1. Challenges in Addressing Psychosocial Needs

Assessing and addressing the psychosocial needs of patients with cancer is crucial for coping, adjustment, and overall well-being [16]. Effective intervention requires the systematic identification of risk factors, symptoms, and associated impacts on medical care to anticipate challenges and mobilize supportive care resources before concerns escalate [17,18,19]. Many centers have begun to integrate psychosocial screening and routine check-ins using PROs into care models to assess patients before, during, and after treatment in accordance with practice standards [20,21,22]. However, there remain gaps in terms of effectively connecting screening, intervention, and outcomes.

### 3.2. Giving Adolescents a Voice

A cancer diagnosis during adolescence, a pivotal time of development for self-discovery, identity formation, and independence, is devastating. It is essential to recognize the importance of adolescents having a voice in their cancer care [23]. However, in the face of uncertainty, many adolescents may not have the skills, resources, or understanding to confidently express their concerns [24]. By building trusting relationships and by actively listening, health professionals can build trusting foundations for such relationships.

### 3.3. Considering the Caregiver’s Perspective

In valuing all voices, the caregiver’s perspective is also important, as their perception of care, participation in care, and child’s health outcomes are indelibly interconnected. Pediatric caregivers, unlike adult caregivers, are uniquely positioned to mediate between their child and the clinician [25]. In decision-making, caregivers are critical in mitigating the psychosocial impact of a childhood cancer diagnosis, which has an important impact on the child’s health outcomes [26]. Caregiver function and levels of stress/distress have direct implications for the child’s quality of life and psychosocial adjustment outcomes [27,28], and caregiver burden and parents’ quality of life are strongly associated with, and moderated by, the child’s treatment status [29]. Furthermore, the degree to which parents perceive care as being family-centered may influence the degree of caregiver burden and, indirectly, the parents’ adaptation [30]. The mediator and advocate roles compel the active inclusion of the caregiver’s voice in a child’s care [31], and the caregiver perspective can provide valuable insight into the guidance needed for psychosocial models of care [32]. Utilizing validated tools can enhance communication, support, and overall well-being, and is critical to treating children with cancer.

### 3.4. Use of Patient-Reported Outcomes

Structured communication tools, such as PRO measures, can aid communication in complex scenarios [33]. These tools can help patients and caregivers to communicate their concerns and actively participate in their own care and decision-making processes. By asking what matters most and acknowledging everyone’s unique perspective, preferences, and values, healthcare professionals can build trusting relationships and provide better support throughout the cancer experience [34].

Despite the recognized benefits of using PROs to assess subjective symptoms and health status, as well as providing timely and actionable information, uptake in pediatric oncology is disappointingly low in clinical practice and trials [35]. Only 8% of clinical trials have included PROs in the last two decades [36]. There are multiple barriers to the use of PROs, including concerns regarding appropriateness, timing, interpretation, and staffing. Additionally, there are concerns regarding the validity of parent proxy reporting for children who may be too young, too unwell, or unwilling to report themselves [37,38]. Consequently, pediatric oncology has not achieved the same gains made in adult cancers with the integration of PROs in routine practice [39,40]. This is particularly relevant to the gains made in quality indicators (QI) for assessing the quality of care. In adult cancer, several sets of QI measures have been established; however, in pediatric cancer, quality assurance systems and guidelines are missing [39].

While the discrepancies between child and parent proxy reporting of PROs are widely described, the recommendations are to value all voices—those of the child, parent, and clinicians—acknowledging the unique perspective each brings to understanding the effects of cancer on a child [41]. The potential benefits of PROs in pediatric oncology will only be realized by efforts to overcome barriers to ensure that children and families are included in decision-making processes and that the outcomes that matter most to them are prioritized [42]. Guidelines on how to implement, interpret, and act upon PROs in children’s cancer while considering these challenges are urgently needed. Having these guidelines may help to establish programs that bridge any gaps leading to palliative care.

## 4. Recognizing the Benefit of Palliative Care

Effective pediatric palliative care (PPC) programs build partnerships between patients, caregivers, and providers to address medical, psychosocial, and spiritual needs during one’s cancer trajectory and at the end of life [43,44]. Clear communication, shared decision making, and responsiveness to articulated needs has been critical for trust-building across stakeholders and improving family-centered care [45,46]. Whether or not the patient survives, the involvement of the PPC team can be beneficial for pain management, patient and family education, and support for critical medical decisions [47]. In some settings, within certain treatment populations (e.g., stem cell transplant), established protocols for PPC referrals either require every patient to receive an initial assessment or identify patients experiencing certain situational “triggers”, such as high-risk disease, organ dysfunction, second transplant, or pediatric sibling donors, to receive an automatic escalation in supportive care [48]. Nevertheless, many families of patients in low- and middle-income countries are not given the opportunity to participate in discussions to determine goals of care or how to prioritize the quality of life and focus care on the relief of pain and discomfort for their child [49,50]. Much of the available literature conveys significant benefits to patient and family well-being when PPC is introduced early in care, and increases as appropriate based on medical status, prognosis, and patient and family priorities [51,52].

## 5. Survivorship and Life after Cancer Therapy

Young cancer survivors have a lifetime to endure the consequences of cancer and its treatment. Long-term health risks including cardiomyopathy, respiratory distress, musculoskeletal problems, endocrine dysfunction, and neurological impairment, as well as secondary neoplasms, which are common [53]. As a result, the risk of early mortality due to late effects of cancer is higher for survivors diagnosed during childhood compared to those diagnosed after the age of 20 years [53]. 

Over the past decade, many national/regional groups and cancer institutions have initiated large prospective cohort studies and registries to characterize the health and psychosocial outcomes of childhood cancer survivors. These chronic consequences of childhood cancer treatment negatively impact the physical and psychosocial wellbeing of pediatric cancer survivors and emphasize cancer survivorship as an unmet need [54,55]. However, it is important to note that these registries reflect the effects of conventional anti-cancer therapies (chemotherapy/radiotherapy) and may not reflect the trajectories of survivorship comorbidities associated with novel anti-cancer therapies, such as immunotherapy. The collection and assessment of this information will take decades, and represents an urgent area of need moving forward.

Surveillance and late effects screening may allow for the detection of health problems at early stages, when they are more amenable to treatment. As such, many international pediatric oncology groups have recommended evidence-based systematic screening for late effects in survivors with childhood cancer. For example, the International Late Effects of Childhood Cancer Guideline Harmonization Group has proposed a risk-based, exposure-related CPG for the screening and management of late effects in survivors [56]. “Risk-based” or “exposure-based” care refers to a personalized, systematic plan of regular screening, surveillance, and prevention strategies to detect recurrence and late effects according to patient-specific risk factors and therapeutic exposures. Such focused implementation strategies will be crucial to improving surveillance care for long-term survivors of childhood cancer. Similarly, cancer and its treatment lead to interruptions in expected developmental milestones, such as completing school and establishing independence, and the impacts of cancer can leave a young survivor grappling with complex psychosocial concerns [57]. There are recommendations for regular mental health assessment and swift referral for support, but the current models of survivorship care often focus solely on the detection of recurrence, neglecting these broader long-term needs [50,57]. While large cohort studies of survivors have highlighted impacts on health-related quality of life, cognitive and functional outcomes, fertility and sexual health, work, and education, there remains scant evidence regarding how to address these problems. Health behaviors, self-efficacy, and self-management likely play key roles and should be studied further [58,59]. Optimal cancer survivorship programs should also comprise health promotion activities, specialty referrals, and psychosocial interventions to address all domains of well-being and quality of life for both high-income and low-income countries.

## 6. Interventions, Models of Care, and Supportive Care Plans

More than half of childhood and young adult survivors express a lack of awareness that cancer treatments can induce future side effects or health difficulties [60], with the majority unable to name a single medication they received during their treatment [61]. This highlights the challenges in placing the burden of navigating survivorship on patients and suggests that a care-team-led, proactive approach to survivorship planning may be more effective. However, many survivors report minimal confidence in the ability of primary care physicians to manage their survivorship-related concerns, which is likely related to a lack of specific training for primary care physicians to manage the complex comorbidities that survivors experience. In fact, only 30% of primary clinicians express being “very comfortable” providing survivorship care, despite the survivorship care plan (SCP) provided by oncologists when survivors are discharged to primary care [62,62]. This highlights the challenging paradox of survivorship care and where the responsibility for providing care resides.

The provision of personalized health risk counseling has been increasingly recognized as an imperative component of pediatric cancer survivorship programs. Such individualized education aims to promote the age-appropriate ownership of childhood survivors’ health and active participation during survivorship through the collaborative development of a SCP [63]. The “survivor healthcare passport” is an intervention designed to improve long-term adherence with surveillance for long-term adverse effects, and to meet the unmet medical information needs of survivors [64]. It is derived from an individual’s diagnosis, treatment history, treatment-associated health risks, and follow-up recommendations. Although there is questionable evidence on the effectiveness of SCPs in improving health outcomes in the adult cancer population [65,65], studies on childhood cancer survivors have suggested that SCPs could effectively improve survivors’ decision-making processes regarding their future health care and adherence to risk-based surveillance for late effects [66,67]. Thus, there may be an increase in long-term surveillance for late effects of childhood cancer following the implementation of user-friendly and individualized care plans. Telehealth interventions are also emerging as a promising strategy for delivering valuable information to young cancer survivors. Recent studies have demonstrated the acceptability and feasibility of delivering SCPs through web-based and mobile applications [68,69]. Future work should include evaluating novel techniques and delivery formats of health education for survivors of childhood cancer, as well as its impact on distal outcomes, such as lifestyle modification, uptake of screening practices, and cost-effectiveness.

While supportive care and cancer survivorship have improved, there remains a lack of pathobiological knowledge of these adverse side effects to implement targeted treatment strategies. The heterogeneous nature of these long-term conditions indicates significant endogenous (i.e., genetic) variables at play; however, the lack of etiologic research findings, particularly in pediatric cohorts, precludes the development of effective treatment measures. Methods to prevent or optimally manage these complications early in their etiology are scarce, and are often borrowed from similar, unrelated conditions. In addition to efforts to optimize the identification and management of late effects, translational and preclinical research initiatives are critical to generating the knowledge that is needed to understand these complications and to identify methods to mitigate and prevent the collateral damage of cancer treatment.

## 7. Leveraging Technology in Supportive Care

Generating evidence on the use, efficacy, and safety of cancer therapy is critical. While randomized controlled trials (RCTs) are considered the gold standard, they are performed in specific patient populations under controlled conditions, limiting their generalizability and the transportability of their findings. When combined with rigorous design and analytic methods, real-world data (RWD) can complement RCT data. Proposed applications for RWD include: (1) following RCTs to refine efficacy estimates, including in underrepresented subgroups; (2) defining effectiveness and comparative efficacy in settings where RCTs are impracticable due to small numbers, including in pediatric oncology [70]; and (3) enhancing our understanding of safety, especially regarding the late adverse events that occur after a standard RCT observation period [71].

Given the fragmentation of healthcare delivery globally, there are a range of RWD types that can be used in isolation and in tandem to address important clinical questions. In broad, categories these are described in Table 2, and include: 

***Registry data:*** Efforts to collect uniform and systematic data on specific cohorts can provide high-quality observational data, and are a well-established research data source. Examples include the National Cancer Institute Surveillance, Epidemiology, and End Results program; the Center for International Blood and Marrow Transplantation Research; and the CONCORD Program for global surveillance of cancer survival. While these datasets provide high-fidelity data, substantial person-time and effort is required in order to establish, maintain, and update them.***Administrative or claims data:*** Health administrative data, such as data submitted for reimbursement related to diagnoses and services rendered during patient visits and hospitalizations, can be compiled longitudinally into datasets. Examples that have been leveraged for research include the Pediatric Health Information System and Medicare Benefits Schedule [72].***Electronic health record*** (***EHR***)***:*** Mining the EHRs for RWD holds enormous potential due to the highly granular data. However, data are collected for clinical purposes and may be stored as unstructured text, limiting their usability without advanced data science support. Data are often limited to single institutions, although consortia are being developed by investigators and healthcare aggregators [73,74,75].***Healthcare aggregators/health technology data companies:*** Combining data across varied clinical sources and sites using a common patient identifier has been performed by commercial and non-profit entities, and may help to overcome the disparate nature of healthcare delivery. Sentinel, OptumLabs, and Flatiron Health are established healthcare aggregators, but pediatric patients are underrepresented, and equivalent resources in this space are lacking.

To maximize the use of RWD, data linkage must be employed. Data linkage aims to combine diverse sources of data regarding an individual to create a more comprehensive understanding of their life experiences, events, and/or interventions. Local laws and regulations govern access to and ability to link types of data. Examples of successful data linkage in practice include the supplementation of clinical trial data with administrative data [76,77], the linking of pathology and clinical data for patients with melanoma data to measure quality [78], and the linkage of cancer registry data and health administrative data to enhance the reporting of cancer outcomes [79,80]. The sources of information and the identifiers used to link data (e.g., date of birth, hospital number) will influence the complexity of the data linkage. Further, data recorded for clinical purposes may not represent what is needed in order to answer additional clinical questions. To overcome this, multiple data sources may need to be linked to obtain all the information needed. However, data quality can be impacted by the identifiers and linkage methods which are utilized, coding errors in source documents, and missing data [81,82,83]. There is a need for investment in data infrastructure to further develop and improve data linkage to enhance supportive care [84].

Efforts are ongoing to link clinical trial data with EHR data extraction to improve the accuracy of data capture [73,75,85]. One example is ExtractEHR, a tool using code to automatically extract and process EHR data to answer clinical questions. ExtractEHR has accurately identified adverse events experienced during chemotherapy and provided a more comprehensive assessment of patients’ experiences during therapy than was obtained using clinical trial data alone [73,85]. The automated ascertainment of data is being tested in ongoing clinical trials to link EHR data collected for patient-care purposes to data collected for clinical trial reporting, which may provide accurate and comprehensive data with which to tailor supportive care.

## 8. Fundamental General Pediatric Care Should Remain Embedded in Oncology

Pediatric cancer patients and their parents report feeling disconnected from their primary care providers during cancer treatment [86]. Anecdotally, patients and their families are frequently told to view their oncologists as their primary provider during therapy; however, no published reports offer guidance for routine pediatric screening through oncology clinics. Therefore, routine pediatric screening needs to be explored and considered in pediatric oncologic care where appropriate, as the medical homes for many children resides within oncology units for months to years. This practice, of course, varies by location and should be adapted according to region.

Embedding general pediatric care in oncology has two primary benefits. Firstly, oncologists rarely complete developmental screening for young patients, unlike the developmental surveillance typically performed at health supervision visits [87]. More routine developmental screens should potentially be incorporated as well, like psychosocial screening, as described previously. While theoretical risk exists for cancer therapy-related stress and side effects to cause false positives when using general pediatric screening tools, this risk must be balanced against not catching and intervening in developmental delays if no screening is performed for months to years while the patient is under oncological care. Secondly, primary care physicians report feeling unprepared to care for cancer survivors [86,88], despite playing a critical role after cancer treatment [89,90]. Given the documented struggles of screening while receiving therapy from oncologists and the struggles of primary physicians to provide care in survivorship, a more holistic, multidisciplinary approach to integrating general pediatric care into oncology clinics is imperative. For example, re-vaccination after therapy is completed is often required, and there are few practices comfortable with providing vaccines to patients treated for cancer or blood disorders [91,92,93]. As re-vaccination has been proven to be feasible and more timely if performed through an oncology clinic [92], opening embedded primary care/vaccination clinics within oncology centers ought to be considered.

### 8.1. Advances in Approaches to and Optimization of Nutrition

Childhood cancer treatment has significant impacts on nutritional status. In addition, nutritional status is altered by social determinants of health, such as food security and psychosocial determinants that affect food preparation and intake [94]. During treatment, food insecurity is the most common household material hardship, impacting approximately one in five families [95]. The quality and quantity of food and nutritional supplements consumed, in the past or in the present, may affect the pathogenesis and biology of the cancer’s response to treatment side effects and quality of life [96,97,98]. Addressing these critical determinants of food availability and how to best prepare food is an important focus of nutritional oncology.

### 8.2. Nutritional Status

Malnutrition and overnutrition are common in children with cancer, occurring in up to 70% and 25–75%, respectively [99]. Malnutrition can result in increased infections, poor tolerance, delays in treatment, and organ dysfunction, among other comorbidities. In addition, overnutrition along with higher BMI has been found to be significantly associated with increased mortality rate, worse event-free survival, and a trend towards greater risk of relapse in children with ALL [100]. Despite the critical impact of nutritional status in children along the cancer continuum, nutritional screenings and assessments remain unstandardized and institution-specific [101]. Assessment tools such as weight, height, and BMI are unable to distinguish muscle from adipose tissue [102], whereas more reliable indicators of nutritional status, such as Mid-Upper Arm Circumference, are not frequently adopted [101,103]. The incorporation of nutritional approaches and comprehensive, systematic screening are essential for optimizing outcomes in children with cancer and in survivors. Nutritional interventions are necessary not only avoid malnutrition, but also to support optimal growth and development.

### 8.3. Improved Dietary Intake

According to the World Cancer Research Fund (WCRF), one-third of the most common adult cancers can be prevented by a healthy diet, being physically active, and maintaining a normal body weight [104]. Children suffering from being overweight often suffer from obesity as adults; this is compounded by the known risk of metabolic syndrome in survivors of childhood cancer [105,106,107,108]. In addition, poorly managed treatment side effects can lead to long-term poor dietary behaviors. The healthy dietary behaviors of children with cancer often plummet after diagnosis, with decreased intake of fruit and vegetables, increase in processed “junk” foods, and overall larger portion sizes [105,109]. It is critical that oncologists promote healthy eating patterns and behaviors in patients, survivors, and caregivers. That said, this must be carried out while understanding patient constraints and symptom burden, and age-appropriate advice which speaks to the need for understanding and incorporating the developmental milestones into routine oncologic care must be provided [103]. Partnering with an oncology dietician is crucial to achieving these aims. 

In addition to a clinical focus on healthy eating, additional research is needed in order to enhance our understanding of the impacts of nutrition on cancer before, during, and after therapy. The field of nutritional oncology is burgeoning, with multi-center trials needed in order to understand the mechanisms of nutritional morbidities in cancer pathogenesis and outcomes. The establishment of standardized nutritional assessments and interventions is also necessary. 

### 8.4. Exercise

While the physical inactivity of AYAs with cancer approximates that of healthy AYAs, nearly 25% of AYA survivors are completely sedentary [109,110], engaging in lower physical activity levels than their healthy peers [111]. This sedentary behavior results in a diminished quality of life, as well as symptomology such as fatigue, lower muscle mass, and weakness, similar to adults aged > 65 [112]. In addition, during cancer treatment, oncologists should aim to promote physical activity whenever it is feasible for the patient, the given benefits of improved physical functioning, body composition, immune and cardiorespiratory symptoms, energy, sleep, and health-related quality of life (HRQOL) [113]. After cancer therapy, physical activity has the potential to reduce treatment-related late effects by improving fatigue, depressive symptoms, cognitive performance, and psychological well-being [111,114,115,116,117], as well as lowering future cancer risks. The International Pediatric Oncology Exercise Guidelines, developed by 122 individuals from 21 countries, reveal a collective agreement that children and adolescents with cancer need to “move more” [118]. The research on exercise oncology is blossoming, but gaps in the field make it difficult for clinicians to make clear-cut recommendations. Research needs to focus on utilizing direct, objective measures of physical activity, not simply self-reporting, with the inclusion of not only aerobic interventions, but also strength training [119]. Additional studies utilizing objective measures of body composition and strength are vital. In addition, a clear understanding of appropriate timing within the cancer continuum and preferred programs for different age groups, including those on and off therapy, are needed.

## 9. Conclusions and Future Directions

The health and optimization of outcomes for pediatric cancer patients relies on the successful advancement of supportive care to ease the treatment burden and mitigate long-term treatment impacts. The future of advancement in pediatric supportive care relies on the successful integration of strategies, as well as prioritizing the research, implementation, and development of innovations in these areas. As with most pediatric oncologic care, while there is a need for personalized or precision approaches to the patient, we should ensure that these are based on evidence where applicable and guarantee CPG-consistent approaches. By incorporating technology and datasets, we can seek answers even if a clinical trial is not feasible. Through earlier integration of supportive care and hearing the patients’ and caregivers’ voices through PROs, we can ensure that their contributions and experiences are heard and that their care plans incorporate these critical aspects. Finally, we must remember that children with cancer are first children, and elements of general and developmental pediatrics like growth, nutrition, development, and physical activity must be integrated into care for this vulnerable population. While available resources vary across the globe, these priorities can and should be modified to fit specific contexts and clinical practice settings. Increased research funding for pediatric supportive care is critical to addressing equality and disparities in these efforts. Our patients deserve to grow and live healthy, productive lives with optimized, enriched supportive care throughout the spectrum of cancer care, from diagnosis to survivorship. Enhanced supportive care the offers potential for an enhanced outcome, optimized quality of life, and a meaningful future for our patients. 

## Figures and Tables

**Table 1 cancers-15-05549-t001:** Opportunities and critical areas of development for optimized pediatric supportive Care.

Development, Use, and Implementation of Clinical Practice Guidelines
Empowering patients and families and giving children a voice in their care
Addressing psychosocial factors and the needs of patients and families
Using patient-reported outcomes routinely
Recognizing the benefit of and actively incorporating early palliative care
Optimizing survivorship, reducing late effects, and promoting survivorship research
Leveraging technology to study and improve supportive care
Re-embedding general pediatric and developmental care into oncologic care

**Table 2 cancers-15-05549-t002:** Real-world data types that can be leveraged to study and improve supportive care.

**Registry Data**	National Cancer Institute Surveillance, Epidemiology, and End Results (SEER) Program; Center for International Blood and Marrow Transplantation Research (CIBMTR); CONCORD Program for Global Surveillance of Cancer Survival
**Administrative or Claims data**	Pediatric Health Information System (PHIS) and Medicare Benefits Data
**Electronic health record**	Institution based medical record systems (e.g., EPIC, Cerner, All-Scripts)
**Healthcare aggregators/health technology data companies**	Sentinel, OptumLabs, and Flatiron Health

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
