# Peer review of "Supportive Care in Pediatric Oncology: Opportunities and Future Directions"

_cancers, 2023, doi:10.3390/cancers15235549_

Round 1
Reviewer 1 Report
Comments and Suggestions for Authors
Overview:
In this manuscript, “Supportive Care in Pediatric Oncology: Opportunities & Future Directions,” the authors aim to describe the importance of standardizing supportive care practices in pediatric oncology and incorporating the patient and family experience. I applaud the authors for taking on this important and salient topic. It is clear that the authors dedicated a significant amount of time to conducting and synthesizing the material in this perspective piece, and I think that their findings have the potential to be a meaningful addition to the literature.
However, as written, this manuscript needs editing to enhance the clarity of the message and narrative flow. The ideas presented by the authors are interesting and valuable to the field of pediatric palliative care, but their impact is lessened by the writing style. Additionally, the authors should address a number of questions and concerns (delineated below) to improve the content and quality of the manuscript.
Section 2 Overall:
This nicely sets the stage for the role CPGs in pediatric oncology. To clarify, Line 72 is incorrect. The first CPG was actually published in 2012 and was for febrile neutropenia. Please see following reference:
Lehrnbecher T, Phillips R, Alexander S, Alvaro F, Carlesse F, Fisher B et al (2012) Guideline for the management of fever and neutropenia in children with cancer and/or undergoing hematopoietic stem-cell transplantation. J Clin Oncol 30:4427–4438
Section 2.1:
I worry about the first sentence. It brings about a negative connotation that may cause a reader to feel blame for CPGs not being well implemented. I would either find a good citation for line 81 or remove and begin this section with “For CPGs to be….”
Also, this section is not very well organized. Why is the GRADE system being mentioned in a sub header for CPG implementation? This needs to be either better explained or the layout of this section should be clearer. I.e. important factors for successful CPG implementation include: the GRADE, developer input and stakeholder input.
Section 2.2
Line 98-101 – need a reference here
Section 3 Overall:
Suggest moving section 3.3 to be your first sub header; thus the flow would be why there are challenges in addressing psychosocial needs, adolescent + parent voice importance, followed by PROs – an actionable way to give patients + families a voice.
Section 3.1
Line 123 – Suggest softening the language from “Do not have the skills” to “may not have the skills”.
Section 3.2
Line 127-18 – need a reference for this statement
Section 3.4
Line 164-166 -Can the authors clarify briefly what these benefits in adult cancer are that we should pay attention to it in pediatrics?
I think the last sentence of section 3.4 should be something connecting PROs/supportive care/giving patients and families a voice to palliative care. It feels like a bit of a hard turn to jump into a topic of palliative care
Section 4:
The specific mention of palliative care is incredibly important to overall holistic care, especially in this paper that is highlighting the need for improved supportive care in pediatric oncology. This paper already includes a robust reference section, however for line 182 – I would like to see a broader range of references, specifically ones that don’t silo into the stem cell transplant world. Additionally, line 190’s reference (#46) is specific to Indonesia and therefore the statement is a little too general to use just this isolated reference. Suggest changing the language to use this study from Indonesia as an example that may suggest low-middle income countries have fewer opportunities to participate in discussions to determine goals of care, or to add additional supporting references.
Lines 190-192 should have references.
Section 5:
Line 199 – reference needed at the end of this statement
Lines 202-204 – reference needed here
Lines 213-217 – please specify that this is personalized based on the therapeutic regimen received for a patient’s cancer
Section 6:
Line 270: correct researching to research
Line 272 -274: I don’t think it is correct to state that the pathophysiology of cardiomyopathy or neuropathy from chemotherapy agents in long-term cancer survivors is unknown. This paper provides a nice example of several known mechanisms for chemotherapy induced neuropathy with a plethora of references throughout.
Yoon SY, Oh J. Neuropathic cancer pain: prevalence, pathophysiology, and management. Korean J Intern Med. 2018 Nov;33(6):1058-1069. doi: 10.3904/kjim.2018.162. Epub 2018 Jun 25. PMID: 29929349; PMCID: PMC6234399.
If your intention behind this statement is a different message, I think this sentence needs to be reworked.
Section 7:
This is a very informative section and great for sparking reader interest/ideas surrounding data collection on small patient populations.
Section 8 Overall:
This is an interesting topic regarding general pediatric care while a patient is undergoing cancer treatment. As an oncologist, I agree with the sentiment that the pediatric oncologist absorbs the care of the entire child and that some holistic pieces of care may deserve better integration, specifically the psychosocial aspects. I would argue that developmental screening and “normalcy” is often put on hold during a child’s cancer treatment because some of our treatment modalities, not to mention the stress of the situation, can impact how a child may be viewed on traditional developmental scales. In summary, I don’t think there are validated developmental tools to use to appropriately assess children undergoing active chemotherapy and would be hesitant to insinuate this in this section.
Section 8.1:
Lines 365-367 – reference is needed here
I also steer you towards this article by Dr. Bona at Dana Farber, discussing aspects of material hardship on pediatric oncology patients and families. This could be helpful for referencing purposes as mentioned above, as well as expanding upon this section.
Bona, Kira, et al. "Trajectory of material hardship and income poverty in families of children undergoing chemotherapy: a prospective cohort study." Pediatric blood & cancer 63.1 (2016): 105-111.
Section 8.3:
Line 389-391 - reference needed
Line 397-398 – beware of changes in font size. Make sure consistent throughout text
Section 8.4:
It is unclear what the value of this sub-section is. Your first sentence highlights the need for increased research, which is fair, however your second sentence could very easily be interpreted as offensive to oncologists and does not entirely serve your paper. The third sentence is essentially a repeat of sentence 1. I would encourage you to eliminate this section and integrate the sentiment of increase research surrounding diet and oncology somewhere else.
Section 8.5:
Lines 416-418 - This is great in theory, but to achieve may be quite different due to side effects of therapy and thus it is recommended to soften this sentence or acknowledge that this may not always be possible, regardless of the health benefits.
Lines 421-424 – this needs a reference. I also question the validity of this statement as many clinical trial documentation and physical exams include a performance score, which could be representative of physical activeness
I love the support and acknowledgement of different types of physical activities like strength training.
Section 11
I think this section is inappropriately numbered. You go from 8 to 11.
A nice conclusion.
Comments on the Quality of English Language
Minor corrections needed. See above.
Author Response
REVIEWER 1
Section 2 Overall:
This nicely sets the stage for the role CPGs in pediatric oncology. To clarify, Line 72 is incorrect. The first CPG was actually published in 2012 and was for febrile neutropenia. Please see following reference:
Lehrnbecher T, Phillips R, Alexander S, Alvaro F, Carlesse F, Fisher B et al (2012) Guideline for the management of fever and neutropenia in children with cancer and/or undergoing hematopoietic stem-cell transplantation. J Clin Oncol 30:4427–4438
Response: Thank you for pointing this out. The first published pediatric supportive care CPG (2011) addressed the chemotherapy emetogenicity classification. The citation has been corrected.
Section 2.1:
I worry about the first sentence. It brings about a negative connotation that may cause a reader to feel blame for CPGs not being well implemented. I would either find a good citation for line 81 or remove and begin this section with “For CPGs to be….”
Response: Thank you for this caution, we have removed this sentence as we do not to want to stimulate any negative connotations.
Also, this section is not very well organized. Why is the GRADE system being mentioned in a sub header for CPG implementation? This needs to be either better explained or the layout of this section should be clearer. I.e. important factors for successful CPG implementation include: the GRADE, developer input and stakeholder input.
Response: Thank you for these suggestions. This paragraph has been revised to emphasize important factors for successful CPG implementation.
Section 2.2
Line 98-101 – need a reference here
Response: We have now cited 2 papers that describe gaps in the availability of CPGs on various supportive care topics that are important in pediatric oncology.
Section 3 Overall:
Suggest moving section 3.3 to be your first sub header; thus the flow would be why there are challenges in addressing psychosocial needs, adolescent + parent voice importance, followed by PROs – an actionable way to give patients + families a voice.
Response: We see the value in this suggestion. Accordingly, we have adjusted the flow as suggested so that the text is in the following order: 1) challenges, 2) adolescent voice, 3) parent voice, and 4) PROs.
Section 3.1
Line 123 – Suggest softening the language from “Do not have the skills” to “may not have the skills”.
Response: This language has been revised as suggested.
Section 3.2
Line 127-18 – need a reference for this statement
Response: A reference has been provided.
Section 3.4
Line 164-166 -Can the authors clarify briefly what these benefits in adult cancer are that we should pay attention to it in pediatrics?
Response: Yes, this is a good suggestion. An example of a benefit has been provided.
I think the last sentence of section 3.4 should be something connecting PROs/supportive care/giving patients and families a voice to palliative care. It feels like a bit of a hard turn to jump into a topic of palliative care
Response: A transition sentence has been added as suggested.
Section 4:
The specific mention of palliative care is incredibly important to overall holistic care, especially in this paper that is highlighting the need for improved supportive care in pediatric oncology. This paper already includes a robust reference section, however for line 182 – I would like to see a broader range of references, specifically ones that don’t silo into the stem cell transplant world. Additionally, line 190’s reference (#46) is specific to Indonesia and therefore the statement is a little too general to use just this isolated reference. Suggest changing the language to use this study from Indonesia as an example that may suggest low-middle income countries have fewer opportunities to participate in discussions to determine goals of care, or to add additional supporting references.
Response: Many thanks for the feedback. Added a broader range of references outside of SCT, and re-arranged the Indonesia article’s placement to better support reference to low-middle income countries later in the section.
Lines 190-192 should have references.
Response: We have added the requested references and appreciate the suggestions.
Section 5:
Line 199 – reference needed at the end of this statement
Response: We have added the references. Thank you for catching this.
Lines 202-204 – reference needed here
Responses: We have added several references here.
Lines 213-217 – please specify that this is personalized based on the therapeutic regimen received for a patient’s cancer
Response: Thank you for the suggestion. We have edited the sentence to:
“Risk-based” or “exposure-based” care refers to a personalized, systematic plan of regular screening, surveillance, and prevention strategies to detect recurrence and late effects according to patient-specific risk factors and therapeutic exposures.”
Section 6:
Line 270: correct researching to research
Response: Thank you. This has been corrected.
Line 272 -274: I don’t think it is correct to state that the pathophysiology of cardiomyopathy or neuropathy from chemotherapy agents in long-term cancer survivors is unknown. This paper provides a nice example of several known mechanisms for chemotherapy induced neuropathy with a plethora of references throughout.
Yoon SY, Oh J. Neuropathic cancer pain: prevalence, pathophysiology, and management. Korean J Intern Med. 2018 Nov;33(6):1058-1069. doi: 10.3904/kjim.2018.162. Epub 2018 Jun 25. PMID: 29929349; PMCID: PMC6234399.
If your intention behind this statement is a different message, I think this sentence needs to be reworked.
Response: You are correct. It was not meant that way and the sentence has been removed so as not to cause readers any confusion.
Section 7:
This is a very informative section and great for sparking reader interest/ideas surrounding data collection on small patient populations.
Response: We thank the reviewers for their kind words.
Section 8 Overall:
This is an interesting topic regarding general pediatric care while a patient is undergoing cancer treatment. As an oncologist, I agree with the sentiment that the pediatric oncologist absorbs the care of the entire child and that some holistic pieces of care may deserve better integration, specifically the psychosocial aspects. I would argue that developmental screening and “normalcy” is often put on hold during a child’s cancer treatment because some of our treatment modalities, not to mention the stress of the situation, can impact how a child may be viewed on traditional developmental scales. In summary, I don’t think there are validated developmental tools to use to appropriately assess children undergoing active chemotherapy and would be hesitant to insinuate this in this sectionResponse: This section has been softened. We do not mean to imply there are validated developmental tools specifically for oncology patients. We feel the risk of delays in diagnosis for developmental conditions outweighs the risk of false positive screening results for general pediatric screens being deployed for cancer patients. Clarifying text has been added.
Section 8.1:
Lines 365-367 – reference is needed here
I also steer you towards this article by Dr. Bona at Dana Farber, discussing aspects of material hardship on pediatric oncology patients and families. This could be helpful for referencing purposes as mentioned above, as well as expanding upon this section.
Bona, Kira, et al. "Trajectory of material hardship and income poverty in families of children undergoing chemotherapy: a prospective cohort study." Pediatric blood & cancer 63.1 (2016): 105-111.
Response: this is an excellent point. The Bona reference has been added to an expanded paragraph in section 8.1.
Section 8.3:
Line 389-391 - reference needed
Response: Agree. Reference added.
Line 397-398 – beware of changes in font size. Make sure consistent throughout text
Response: Agree but these errors are from the Editor/Journal fontsetter, not our submission. We will be sure to have them correct these throughout the document.
Section 8.4:
It is unclear what the value of this sub-section is. Your first sentence highlights the need for increased research, which is fair, however your second sentence could very easily be interpreted as offensive to oncologists and does not entirely serve your paper. The third sentence is essentially a repeat of sentence 1. I would encourage you to eliminate this section and integrate the sentiment of increase research surrounding diet and oncology somewhere else.
Response: This section has been removed.
Section 8.5:
Lines 416-418 - This is great in theory, but to achieve may be quite different due to side effects of therapy and thus it is recommended to soften this sentence or acknowledge that this may not always be possible, regardless of the health benefits.
Response: We softened the sentence and acknowledged that exercise may not always be possible.
Lines 421-424 – this needs a reference. I also question the validity of this statement as many clinical trial documentation and physical exams include a performance score, which could be representative of physical activeness
Response: Sentence has been removed.
I love the support and acknowledgement of different types of physical activities like strength training.
Response: Thank you!
Section 11
I think this section is inappropriately numbered. You go from 8 to 11.
Response: Yes, this was a typographical error done by the editor, not us. We have fixed it to 9.
A nice conclusion.
Response: Thank you!
Reviewer 2 Report
Comments and Suggestions for Authors
thank you for submitting this plea for excellent supportive care
few comments;
1 You emphasize in the title supportive care in paediatric oncology which covers from diagnosis till 5 years after treatment, after that follows survivorship. If you want to cover the whole field of survivorship as well as you do in your manuscript I would also like to see that In the title
I would also find it worthwhile to restrict the manuscript to the first 5 years as you then emphasize the supportive care expertise in pediatric oncology
2; Because you emphasize patient/parent reported outcomes I would also stress the findings of the adults that using these tools definitely leads to better survival JAMA July 11, 2017 Volume 318, Number 2 Ethan Basch, MD,MSc For instance add it in line 54 page 2, this is an extremely important finding to improve supportive care in all its aspects in paediatric oncology
And stress in the introduction that treatment related morbidity is high especially in the haem-onc patients with infectious morbidity (with references) That makes our need to give excellent supportive care higher and makes your manuscript stronger
2.1. Clinical Practice Guideline (CPG) Implementation
Emphasize as well the necessary adaptation in resourse limited countries, within the organizations SIOP and MASCC facilitating the actions necessary to do this adaptation
4. Recognizing the Benefit of Palliative Care
My same comment as with survivor ship. If you include palliative care then mention it in the title and in the introduction or explain to the readers that palliative and supportive care are one and the same for the best way to offer the best quality of life to the patient
5 5. Survivorship and Life After Cancer Therapy
See my comment earlier. Personally I would stick to supportive care diagnosis to first 5 years after diagnosis
8. Fundamental General Pediatric Care Should Remain Embedded in Oncology
I believe this is very different in the different regions of the world, therefore it cannot be generalized
In the first 5 years of treatment and follow up I believe a pediatric oncologist who sees the child most often is in charge to deal with nutrition, endocrine problems but also developmental and physical disabilities. And from the primary center the best care can be offered closer to the childs home so that will mean that the care can be continued there also after those 5 years.
I would suggest to decrease the length of this topic and try to write it more generalized and focused to the period you want to emphasize
Last sentence Our patients deserve to grow and live healthy, productive lives with optimized, 451 enriched supportive care throughout the spectrum of cancer care from diagnosis to survi vorship.
I would again emphasize that by improving supportive care we will decrease mobidity and mortality in the child with cancer and hopefully increase survivalrate.
Author Response
REVIEWER 2
1) You emphasize in the title supportive care in paediatric oncology which covers from diagnosis till 5 years after treatment, after that follows survivorship. If you want to cover the whole field of survivorship as well as you do in your manuscript I would also like to see that In the title
I would also find it worthwhile to restrict the manuscript to the first 5 years as you then emphasize the supportive care expertise in pediatric oncology
Response: Thank you for this comment. The aim of this review is to provide directions for future research in pediatric oncology and in doing so, to focus on clinical and research gaps throughout the cancer care continuum, which critically and fundamentally includes long-term survivorship. Unlike adult oncology, cancer survivorship is linked and part of supportive care, especially within consortia such as MASCC (Multinational Association of Supportive Care in Cancer). Respectfully, we propose to keep this section and title as is to provide readers with a holistic view of issues around pediatric oncology.
2; Because you emphasize patient/parent reported outcomes I would also stress the findings of the adults that using these tools definitely leads to better survival JAMA July 11, 2017 Volume 318, Number 2 Ethan Basch, MD,MSc For instance add it in line 54 page 2, this is an extremely important finding to improve supportive care in all its aspects in paediatric oncology
Response: Have added some text to indicate that great strides in adult supportive care have led the way for pediatrics to follow suit and added several citations within the text under section 3.4.
2.1. Clinical Practice Guideline (CPG) Implementation
Emphasize as well the necessary adaptation in resourse limited countries, within the organizations SIOP and MASCC facilitating the actions necessary to do this adaptation
Response: Thank you for this suggestion. We have added the following sentences to this section“The evidence that informs many supportive care CPGs reflects care provided in highly resourced settings. Transparent CPG adaptation to low or moderately resourced settings is, nevertheless, an important way to encourage the provision of evidence-based supportive care for all pediatric patients regardless of where they receive care.”
- Recognizing the Benefit of Palliative Care
My same comment as with survivor ship. If you include palliative care then mention it in the title and in the introduction or explain to the readers that palliative and supportive care are one and the same for the best way to offer the best quality of life to the patient
Response: As mentioned above, in pediatrics survivorship is embedded in supportive care. Respectfully we would ask to keep the title as is which encompasses the entire cancer trajectory and is a critical piece of supportive care.
5 5. Survivorship and Life After Cancer Therapy
See my comment earlier. Personally I would stick to supportive care diagnosis to first 5 years after diagnosis
Response: Thank you for this comment. The aim of this review is to provide directions for future research in pediatric oncology and in doing so, to focus on clinical and research gaps throughout the cancer care continuum, which critically and fundamentally includes long-term survivorship. Childhood cancer survivorship has now become an emerging area due to improvement in survival rates of childhood cancer in both developed and developing countries/regions. Respectfully, we propose to keep this section to provide readers with a holistic view of issues around pediatric oncology.
- Fundamental General Pediatric Care Should Remain Embedded in Oncology
I believe this is very different in the different regions of the world, therefore it cannot be generalized
Response: You are most correct. We have added verbiage to that point.
In the first 5 years of treatment and follow up I believe a pediatric oncologist who sees the child most often is in charge to deal with nutrition, endocrine problems but also developmental and physical disabilities. And from the primary center the best care can be offered closer to the childs home so that will mean that the care can be continued there also after those 5 years. I would suggest to decrease the length of this topic and try to write it more generalized and focused to the period you want to emphasize.
Response: We have revised some of the wording and shortened this section a touch, but do not want the reader to miss the critical issues of nutrition, exercise and overall care that cancer patients need.
Last sentence Our patients deserve to grow and live healthy, productive lives with optimized, 451 enriched supportive care throughout the spectrum of cancer care from diagnosis to survi vorship. I would again emphasize that by improving supportive care we will decrease mobidity and mortality in the child with cancer and hopefully increase survival rate.
Response: We agree wholeheartedly and added a sentence to this effect.
